# What are the biopsychosocial risk factors associated with pain in postpartum runners? Development of a clinical decision tool

**Shefali Mathur Christopher**[1,2]*, **Chad E. Cook**[3,4,5☯], **Suzanne J. Snodgrass**[2☯]

**1** Department of Physical Therapy Education, Elon University, Elon, NC, United States of America,
**2** Discipline of Physiotherapy, School of Health Sciences, The University of Newcastle, Callaghan, NSW, Australia, **3** Doctor of Physical Therapy Division, Department of Orthopaedics, Duke University School of Medicine, Durham, NC, United States of America, **4** Duke Clinical Research Institute, Duke University, Durham, NC, United States of America, **5** Duke Department of Population Health Sciences, Durham, NC, United States of America

☯ These authors contributed equally to this work.
* schristopher3@elon.edu

## Abstract

### Background

In 2019, a majority of runners participating in running events were female and 49% were of childbearing age. Studies have reported that women are initiating or returning to running after childbirth with up to 35% reporting pain. There are no studies exploring running-related pain or risk factors for this pain after childbirth in runners. Postpartum runners have a variety of biomechanical, musculoskeletal, and physiologic impairments from which to recover from when returning to high impact sports like running, which could influence initiating or returning to running. Therefore, the purpose of this study was to identify risk factors associated with running-related pain in postpartum runners with and without pain. This study also aimed to understand the compounding effects of multiple associative risk factors by developing a clinical decision tool to identify postpartum runners at higher risk for pain.

### Methods

Postpartum runners with at least one child ≤36 months who ran once a week and postpartum runners unable to run because of pain, but identified as runners, were surveyed. Running variables (mileage, time to first postpartum run), postpartum variables (delivery type, breastfeeding, incontinence, sleep, fatigue, depression), and demographic information were collected. Risk factors for running-related pain were analyzed in bivariate regression models. Variables meeting criteria (*P*<0.15) were entered into a multivariate logistic regression model to create a clinical decision tool. The tool identified compounding factors that increased the probability of having running-related pain after childbirth.

### Results

Analyses included 538 postpartum runners; 176 (32.7%) reporting running-related pain. Eleven variables were included in the multivariate model with six retained in the clinical

**Data Availability Statement:** Data attached in supplemental S2 File.

**Funding:** The authors received no specific funding for this work. Participant incentives were sponsored by The University of Newcastle Research Training Program funds.

**Competing interests:** The authors have declared that no competing interests exist.

decision tool: runner type-novice (OR 3.51; 95% CI 1.65, 7.48), postpartum accumulated fatigue score of >19 (OR 2.48; 95% CI 1.44, 4.28), previous running injury (OR 1.95; 95% CI 1.31, 2.91), vaginal delivery (OR 1.63; 95% CI 1.06, 2.50), incontinence (OR 1.95; 95% CI 1.31, 2.84) and <6.8 hours of sleep on average per night (OR 1.89; 95% CI 1.28, 2.78). Having $\geq$ 4 risk factors increased the probability of having running-related pain to 61.2%.

## Conclusion

The results of this study provide a deeper understanding of the risk factors for running-related pain in postpartum runners. With this information, clinicians can monitor and educate postpartum runners initiating or returning to running. Education could include details of risk factors, combinations of factors for pain and strategies to mitigate risks. Coaches can adapt running workload accounting for fatigue and sleep fluctuations to optimize recovery and performance. Future longitudinal studies that follow asymptomatic postpartum women returning to running after childbirth over time should be performed to validate these findings.

## Introduction

Injuries are the most common reason for reductions or termination of recreational and fitive running [1–3]. Injury incidence, specifically overuse-related injury, has been reported to occur in as many as 92.4% of recreational runners [4]. Because of significant psychological and physiological health benefits, it is best to keep runners running. Running is associated with reductions in psychological distress, depression, anxiety, and improves one's self-image, and mood [5], and is also linked with a decreased risk of cardiovascular disease, and assists with weight management [6,7].

Postpartum runners potentially make up a large portion of community runners as a majority of women participating in running events have been documented to be of childbearing age [8]. In one study of 406 postpartum runners, 45% returned to running within 4 weeks after childbirth, and 70% returned within 8 weeks [9]. Returning to running after childbirth is challenging for many women [10] as a postpartum woman experiences a plethora of perinatal related musculoskeletal and physiological changes. Early return to running may negatively influence the traditional healing and recovery processes associated with childbirth (e.g., pelvic floor damage, scarring, and strength loss) [11]. Receiving rehabilitative postpartum care is not common practice with respect to timing, intensity, and running [12]. Presently, there are no peer reviewed return to sport protocols guiding postpartum women [12], which may be one of the reasons that 35% of postpartum runners experience running-related pain [10].

A recently published Delphi study had experts report on common musculoskeletal deficiencies and risk factors that contribute to pain in postpartum runners [13]. The included risk factors and deficiencies consisted of strength impairments (abdominal, hip, and pelvic floor weakness, range of motion impairments (hip extension restriction, anterior pelvic tilt, and general hypermobility), flexibility impairments (abdominal wall, and tightness in hip flexors, lumbar extensors, iliotibial band, and hamstrings) and alignment impairments (Trendelenburg sign, dynamic knee valgus, lumbar lordosis, over-pronation, and thoracic kyphosis). The risk factors identified by the expert group for pain in postpartum runners were hip pain, decreased exercise tolerance, pain during pregnancy, trying to exercise "too much too soon", life stressors, and pelvic floor trauma.

Experts, not postpartum runners, proposed the previously described risk factors and impairments. To our knowledge, an in-depth investigation identifying risk factors associated

with running-related pain in postpartum runners has not been published. This study aims to identify potential risk factors associated with pain in postpartum runners, in a case control group of postpartum runners with and without a self-report of running-related pain. This study also aims to explore the compounding effects of multiple associative risk factors by developing a clinical decision tool to identify postpartum runners at higher risk for pain.

## Methods

### Design and reporting standards

An international cross-sectional survey was conducted between December 2019 and January 2021. This web-based, anonymous survey was available through social media and flyers posted in public spaces likely to be frequented by postpartum runners in Durham, Raleigh and Burlington, North Carolina, United States. The flyer was also emailed to physiotherapists colleagues, who treat runners or postpartum women, to share the survey flyer on their social media and post in their clinics all around the United States. This study was approved by university institutional review boards. Before consenting to take the survey, participants were provided with study details (S1 File). Methods and results are reported in accordance with the checklist of reporting results of internet e-surveys (CHERRIES) guidelines [14].

### Survey

As no standardized questionnaires were identified to investigate pain in postpartum runners, this survey was designed utilizing previously published works on postpartum runners [9,10,12,13,15,16]. The survey draft was reviewed by five experts and content was edited. The survey was then piloted by six postpartum runners to test usability and functionality before launching. Pilot data were not included in the statistical analysis.

Participants were women 18-years and older who had given birth to at least one child in the past three years, were running at least one time per week or trying to run but were unable to due to pain, and were not currently pregnant [17,18]. The first three questions of the survey confirmed eligibility and the survey terminated if inclusion criteria were not met (S1 File). Those who were not running at least once a week were provided an additional question to identify the reason for their limitation (pain, time or other), and were included if pain was the limiting factor. The youngest child's date of birth was provided to confirm eligibility.

### Study variables

**Descriptive/Independent variables.** For demographics, age (years), parity (primiparous or multiparous), race (Caucasian or other), education (high school or greater) and relationship status (married yes/no) were collected. For postpartum variables, diastasis recti diagnosis (yes/no), breastfeeding status (yes/no), incontinence (any), delivery type of youngest child (vaginal, cesarean, or other), fatigue (yes/no), postpartum accumulated fatigue scale score (PAFS), Edinburgh postpartum depression score (EDPS), average hours of sleep (hours) and average sleep interruptions (1–5 or more) were collected. To collect running-related variables participants reported on average weekly running amount (miles), time to first postpartum run (weeks), type of runner (e.g., novice, recreational, competitive/elite), and previous running-related injury (RRI). Since the survey spanned the COVID-19 pandemic, in April 2020 a question regarding the pandemic's effect on running mileage was added to the survey.

When possible, validated surveys were used to measure postpartum related variables. Postpartum fatigue, a common postpartum symptom, was measured by asking participants if they experienced fatigue, and those that said yes answered the PAFS. Those that said no, were

coded as 0 for the PAFS score. The PAFS includes questions covering three areas of physical, emotional, and cognitive fatigue and has good validity and internal consistency [19,20]. To measure depression, we used the EPDS, a self-report questionnaire designed to screen new mothers [21] and their emotional experience over the previous seven days [21,22]. It is a widely used screening instrument used for assessing depression and anxiety in the perinatal population [22,23].

**Outcome/Dependent variables.**   In this survey, a report of pain associated with running was the outcome variable. Participants were asked if they had current pain when running, which was scored as "yes" or "no". Although pain descriptors and alleviating/aggravating factors were collected, they were not included in this study.

## Missing values

The raw survey data were evaluated for missing values and there were 12 (0.10%), reflected in a total of 6 (1.11%) cases. Little's test for missingness showed the data were missing completely at random. We performed multiple regression-based imputation to replace missing values and pooled the results of five iterations. Upon completion, all analyses were performed on the pooled imputed dataset.

## Statistical analysis for data modeling

All analyses were performed using SPSS version 26.0.0.2 (IBM corp. Armonk, NY, USA). Descriptive statistics representing raw data for the categories were calculated, including means and standard deviations or proportions and percentages. When appropriate, frequencies and distributions were also calculated. Independent samples t-tests or Chi square tests were performed to understand differences between postpartum runners with and without pain. Before the bivariate analysis, continuous variables were converted to binomial variables using the midpoint of the ROC (Receiver Operating Characteristic) curve generated discrimination threshold.

**Bivariate logistic regression modeling.**   For the initial step of clinical decision tool modeling, we evaluated dedicated cumulative combinations of factors that were related to postpartum running-related pain [24,25]. Bivariate relationships (one predictor to a single outcome) were analyzed with 19 unique logistic regression analyses for the outcome variable (do you currently have pain with running?). When there were multiple variables that measured the same construct (e.g., fatigue and PAFS score or sleep average hours and number of sleep interruptions) the authors identified the single item or scale that most accurately reflected the latent construct. For example, PAFS total score was used for fatigue and for sleep average hours was used for sleep.

**Multivariate logistic regression modeling.**   Variables that achieved a P value of <0.15 in their bivariate logistic regression were retained in the multivariate regression analysis [24]. To ensure appropriate modeling, multicollinearity was assessed for each of the retained variables using Phi and Cramer's V to reflect the data type (nominal). Variables with multicollinearity R values of less than 0.6 were used in the multivariate analysis. For the multivariate analysis, a backward conditional stepwise logistic regression was used [24]. Variables that had 95% confidence intervals that did not cross 1.0 were considered statistically significant.

**Creation of conditions for the clinical decision tool.**   The retained variables in the multivariate model were used to understand the effect of the cumulative combinations of variables on the presence of pain in the sample, a feature typical to clinical decision rules modeling [24]. The retained regression variables from the aforementioned stepwise regression findings were entered into 2x2 contingency tables such that the combination of variables 1 of X, 2 of X and 3

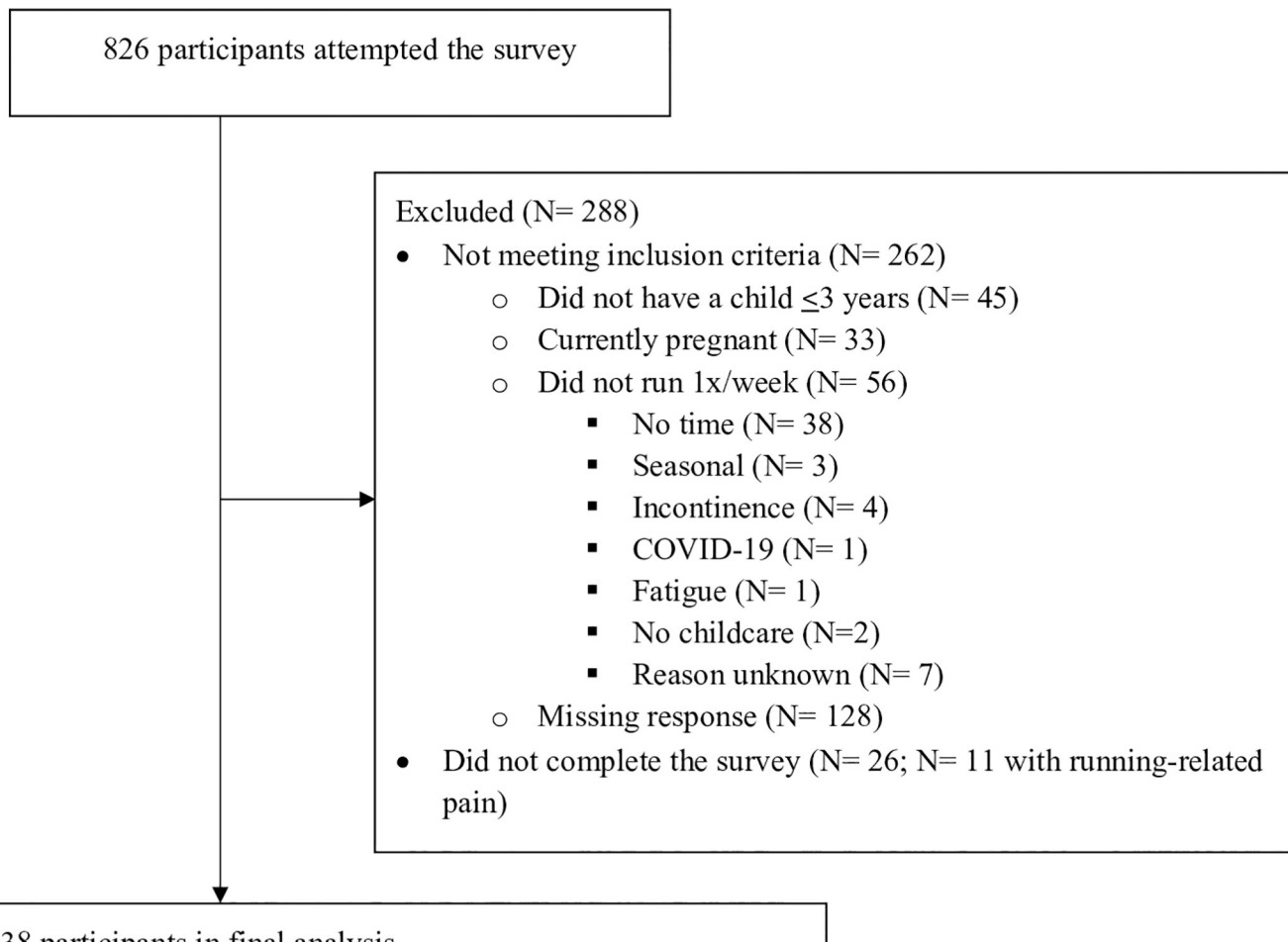

**Fig 1. Flow chart of survey study participants.**

of X and so on generated sensitivity, specificity, positive and negative likelihood ratios and 95% CIs [24]. For each combination (e.g., 1 of X), the odds ratio and 95% confidence intervals, *P* value and Nagelkerke $R^2$ were captured [24,26]. The Nagelkerke $R^2$ is a goodness of fit measure that helps explain the strength of the independent variable with the model [24]. We also include a post-test probability of a negative and positive finding using a post-test prevalence calculator (Diagnostic post-test probability disease calculator) [27].

## Results

### Participants

The survey was initiated by 826 participants; 538 who met the inclusion criteria and completed the survey were included in the final analyses (Fig 1). The majority of incomplete surveys involved women who did not meet the eligibility criteria (n = 262), which triggered the survey to terminate early. Twenty-six eligible respondents did not finish the survey. Eligible non-

completers had similar average weekly running mileage and proportions of each type of runner. The percentage of non-completers with pain (11/26, 42%) was slightly higher than in the completers (176/538, 33%), suggesting the prevalence of running-related pain in postpartum women might be slightly higher than our survey suggests. Information about the demographics of non-completers could not be analyzed as most did not complete that section of the survey.

Among the 538 eligible respondents who completed the survey, 176 (32.7%) reported current pain with running. Postpartum runners in pain had significantly higher prevalence of any incontinence, vaginal deliveries, fatigue, sleep interruptions, novice runners in the group, previous RRI and PAFS and EPDS scores. They also had lower education level, less average weekly hours of sleep, lower total weekly running mileage, when compared to postpartum runners without pain (Table 1).

### Bivariate modeling of associative factors and pain during running

Of the nineteen variables analyzed in the bivariate models, eleven were retained in the multivariate model (Table 2). There were no variables removed for multicollinearity (r values were well below 0.6). When the eleven variables were entered into the multivariate analysis, six variables were retained and were associated with pain with running (Table 3).

### Multivariate modeling of associative factors and pain during running

The sensitivity, specificity, positive and negative likelihood ratios, and probabilities of having pain with running or not (clinical decision tool) in the presence of one or more of the six identified risk factors are outlined in Table 4. The probability of having running-related pain increased with each cumulative risk factor. The use of 4 of 6 to "rule in" the risk for postpartum running pain is recommended for the clinical decision tool, as the confidence intervals of the positive likelihood at this level were narrow and it represented a moderate percentage of the sample (N = 23) with postpartum running-related pain (Table 4).

## Discussion

Women are running after childbirth and up to 35% are reporting pain; however, studies investigating pain in postpartum runners are sparse [10,13]. This study is the first large survey to investigate risk factors for running-related pain in the postpartum population. By creating a clinical decision tool, this study also provides information on the compounding effect of multiple associative variables and pain in postpartum runners. In this sample of runners, the six significant variables associated with having pain were runner type-novice, postpartum accumulated fatigue scale score (>19), previous running injury, most recent delivery-vaginal, incontinence, and average amount of sleep per night (<6.8 hours). When four or more risk factors were present, our model suggested the probability of having pain increased from 32.7% (pre-test prevalence) to 61% (post-test). While this model needs validation in a longitudinal cohort of pain free postpartum runners to determine its predictive capacity, the results provide a deeper understanding of the risk factors for running-related pain in postpartum runners. This may assist health care providers educate postpartum runners and develop interventions that assist postpartum women stay injury free as they initiate or return to running.

### Risk factors associated with postpartum running-related pain

One of the factors that had the highest odds for pain in postpartum runners was self-identifying as a novice runner, a finding that is consistent with previous work [28]. Novice runners are

**Table 1. Descriptive statistics for demographics, postpartum, running and COVID-19 related factors between postpartum runners with (N = 176) and without self-reported running-related pain (n = 362).**

| | Variables | Total (N = 538) | Postpartum pain (N = 176) | Postpartum no pain (N = 362) | P value |
|---|---|---|---|---|---|
| Demographics | Mean age (SD) | 33.62(4.04) | 33.38 (3.81) | 33.74 (3.75) | 0.300 |
| | Parity (1 child) | 237 (44.1) | 84 (47.7) | 153 (42.3) | 0.231 |
| | Race (Caucasian) | 503 (93.5) | 163 (92.6) | 340 (93.91) | 0.564 |
| | Education level (≥High school) | 503 (93.5) | 158 (89.8) | 345 (95.3) | 0.015* |
| | Relationship status (Married) | 526 (97.7) | 170 (96.6) | 356 (98.3) | 0.200 |
| Postpartum | | | | | |
| | Diastasis Recti diagnosis | 115 (21.4) | 46 (26.1) | 69 (19.1) | 0.060 |
| | Breastfeeding/pumping | 252 (46.8) | 85 (48.3) | 167 (46.1) | 0.637 |
| | Incontinence | 230 (42.8) | 95 (54.0) | 135 (37.3) | <0.01* |
| Delivery type | | | | | |
| | Vaginal | 367 (68.2) | 130 (73.9) | 237 (65.5) | 0.050* |
| | Cesarean | 132 (24.5) | 37 (21.0) | 95 (26.2) | 0.187 |
| | VBAC/Vaginal assisted | 39 (7.3) | 9 (5.1) | 30 (8.3) | 0.183 |
| | Fatigue (yes) | 461 (85.7) | 164 (93.2) | 297 (82.0) | 0.001* |
| | Mean Postpartum accumulated fatigue scale (PAFS) score (SD) | 10.54 (8.07) | 13.59 (8.67) | 9.05 (7.33) | <0.001* |
| | Mean Edinburgh Postpartum Depression score (SD) | 6.70 (4.84) | 7.38 (5.07) | 6.37 (4.69) | 0.024* |
| | Mean hours of sleep per night (SD) | 6.67 (1.17) | 6.40 (1.21) | 6.80 (1.12) | <0.001* |
| | Missing values | 1 (0.2) | 1 (0.6) | 0 (0.0) | |
| Number of sleep interruptions per night‡ | | | | | |
| | 0 | 46(8.6) | 11 (6.3) | 35 (9.7) | <0.001* |
| | 1 | 156 (29.0) | 41 (23.30) | 115 (31.77) | |
| | 2 | 168 (31.2) | 59 (33.52) | 109 (30.11) | |
| | 3 | 94 (17.5) | 29 (16.48) | 65 (17.96) | |
| | 4 | 37 (6.9) | 17 (9.66) | 20 (5.52) | |
| | 5 | 15 (2.8) | 3 (1.70) | 12 (3.31) | |
| | >5 | 22 (4.1) | 16 (9.09) | 6 (1.66) | |
| Running | | | | | |
| | Mean total weekly mileage (SD) | 13.07 (12.37) | 11.80 (12.99) | 13.68 (12.02) | 0.100* |
| | Missing values | 1 (0.2) | 1(0.6) | 0 (0.0) | |
| | Mean weeks to first run postpartum(SD) | 12.72(14.31) | 14.53 (16.80) | 11.85 (12.87) | 0.066 |
| | Missing values | 4 (0.7) | 3 (1.7) | 1 (0.3) | |
| Runner type | | | | | |
| | Missing values | 2 (0.4) | 2 (1.1) | 0 (0.0) | |
| | Runner type- Novice | 36 (6.69) | 20 (11.5) | 14 (3.9) | 0.001* |
| | Runner type- Recreational | 401 (74.54) | 126 (72.4) | 275 (76) | 0.380 |
| | Runner type- Elite | 101 (18.77) | 28 (16.1) | 73 (20.2) | 0.260 |
| | Currently running with stroller | 292 (54.2) | 94 (53.4) | 197 (54.4) | 0.826 |
| | Previous running injury | 292 (54.3) | 108 (61.4) | 184 (50.8) | 0.021* |
| COVID-19 | COVID related running changes‡ | 164 (30.3%) | | | |
| | Mileage increased | 64 (11.9) | 20 (35.71) | 44 (40.74) | 0.359 |
| | Mileage decreased | 31 (5.8) | 14 (25%) | 17 (15.74) | |
| | No change in mileage | 69 (12.8) | 22 (39.30) | 47 (43.52) | |

*Significant $P$ <0.05; VBAC–vaginal birth after cesarean

‡- Chi square test; Variables represent number (%) or t-test analysis unless otherwise noted.

**Table 2. Bivariate relationship between risk factors and having current pain with running.**

| | Variable (Binomial distinction) | Odds ratio (95% CI) | P value | Nagelkerke R$^2$ |
|---|---|---|---|---|
| Demographics | Parity (only 1 child) | 1.23 (0.87, 1.79) | 0.232 | 0.004 |
| | Race (Caucasian) | 0.81 (0.40, 1.65) | 0.564 | 0.001 |
| | Education (≥High school) | 0.43 (0.22, 0.86) | 0.017* | 0.014 |
| | Relationship status (Married) | 0.48 (0.15, 1.50) | 0.206 | 0.004 |
| | Diastasis recti diagnosis (yes/no) | 1.50 (0.98, 2.30) | 0.061* | 0.009 |
| Postpartum | Breastfeeding/pumping (yes/no) | 1.09 (0.76, 1.56) | 0.637 | 0.001 |
| | Incontinence: Urine, feces and/or gas (yes/no) | 1.97 (1.37, 2.84) | <0.001* | 0.034 |
| | Delivery type- Vaginal (yes/no) | 1.49 (0.10, 2.22) | 0.050* | 0.010 |
| | Delivery type- C-section (yes/no) | 0.75 (0.49, 1.15) | 0.188 | 0.005 |
| | Delivery type- Other (yes/no) | 0.40 (0.28, 1.29) | 0.187 | 0.005 |
| | PAFS score (≥19) | 2.84 (1.70, 4.73) | <0.001* | 0.041 |
| | EPDS (≥12.5) | 1.84 (1.13, 3.01) | 0.014* | 0.015 |
| | Sleep (≤6.83 hours) | 2.09 (1.45, 3.01) | <0.001* | 0.040 |
| Running | Total weekly mileage (≥15.25 miles) | 0.73 (0.49, 1.10) | 0.134* | 0.006 |
| | First run postpartum (≥24.5 weeks) | 1.61 (0.90, 2.89) | 0.108* | 0.008 |
| | Runner type- Novice (yes/no) | 3.29 (1.62, 6.72) | 0.001* | 0.027 |
| | Runner type- Recreational (yes/no) | 0.77 (0.48, 1.26) | 0.302 | 0.002 |
| | Runner type- Elite (yes/no) | 0.81 (0.54, 1.21) | 0.301 | 0.004 |
| | Previous running injury (yes/no) | 1.54 (1.07, 2.22) | 0.022* | 0.014 |

*Met criteria (p <0.15) for inclusion in multivariate model; Abbreviations; PAFS = Postpartum accumulated fatigue scale; EPDS = Edinburgh Postpartum Depression Score.

more likely to be older, have higher BMI, have had a previous injury, and have no previous experience with running [29]. Novice runners who participate in a self-devised training program are also more likely to be injured compared to those using a structured "couch to 5K" program [27] suggesting the need for a structured program that addresses the potential confounding variables that may increase the risk of injury in a novice runner.

A higher postpartum accumulated fatigue scale (PAFS) score had higher odds for pain in postpartum runners. Fatigue in new parents is well studied, with up to 64% reporting fatigue during the postpartum period [30,31]. In athlete populations, fatigue is often studied to understand underperformance and injury [32]. Insufficient recovery time, the lack of optimal training load, and training intensity have all been associated with fatigue [32,33]. Although fatigue has been well documented in postpartum and athlete populations separately, it has not been

**Table 3. Results of final multivariate model (backwards stepwise) demonstrating variables that are associated with current pain with running postpartum (R$^2$ = 0.161).**

| Variable | Odds Ratio (95% CI) | P value |
|---|---|---|
| Delivery type- Vaginal (yes/no) | 1.63 (1.06, 2.50) | 0.027* |
| Incontinence (Urine, feces and/or gas) | 1.93 (1.31, 2.84) | 0.001* |
| PAFS score (≥19) | 2.48 (1.44, 4.28) | 0.001* |
| Amount of sleep (≤6.84) | 1.89 (1.28, 2.78) | 0.001* |
| Runner type- Novice (yes/no) | 3.51 (1.65, 7.48) | 0.001* |
| Previous running injury (yes/no) | 1.95 (1.31, 2.91) | 0.001* |

*Significant P<0.05; Abbreviations; CI = Confidence intervals; PAFS = Postpartum accumulated fatigue scale.

**Table 4. Clinical prediction tool for postpartum running-related pain based on having different numbers of risk factors and current pain with running.**

| Number (of 6*) risk factors present | Sensitivity (95% CI) | Specificity (95% CI) | Positive Likelihood Ratio (95% CI) | Negative Likelihood Ratio (95% CI) | Post-test probability when finding is positive (%)** | Post-test probability when finding is negative (%) |
|---|---|---|---|---|---|---|
| 1 or more | 98.9 (96.3, 99.8) | 5.0 (3.70, 5.40) | 1.04 (1.0, 1.06) | 0.23 (0.04, 1.00) | 33.6 | 10.1 |
| 2 or more | 87.5 (82.4, 91.6) | 30.10 (27.60, 32.10) | 1.25 (1.14, 1.35) | 0.42 (0.26, 0.64) | 37.8 | 16.9 |
| 3 or more | 63.60 (57.4, 69.5) | 67.4 (64.4, 70.2) | 1.95 (1.61, 2.34) | 0.54 (0.43, 0.66) | 48.7 | 20.8 |
| 4 or more | 23.30 (18.70, 27.40) | 92.80 (90.60, 94.80) | 3.24 (2.01, 5.29) | 0.83 (0.77, 0.90) | 61.2 | 28.7 |
| 5 or more | 5.10 (3.10, 5.70) | 99.7 (98.7, 100.00) | 18.51 (2.46, 390.14) | 0.95 (0.94, 0.98) | 90.0 | 31.6 |
| 6 of 6 | 0.60 (0.00, 0.60) | 100.0 (99.70, 100.00) | Inf (0.12, Inf) | 0.99 (0.99, 1.00) | ~100 | 32.5 |

*Six significant variables: Runner type-novice, postpartum accumulated fatigue scale score (>19), previous running injury, most recent delivery-vaginal, incontinence and amount of sleep (<6.8 hours).

**Pre-test probability was 32.7% before statistical analysis was performed to evaluate cumulative effects of associated variables. Abbreviations; CI = Confidence intervals.

studied in postpartum athlete populations. The finding that postpartum runners with greater accumulated fatigue had higher odds of pain than those with less fatigue may highlight the need to monitor this risk factor in the postpartum running population. Coaches and clinicians may have to adjust typical running workload (frequency, intensity, duration) when working with fatigued postpartum runners. It is important to note however that fatigue is often associated with pain and thus this fatigue-related pain may not be related to running [34]. Therefore, postpartum runners initiating or returning to running should be screened not only for pain but also for fatigue which may also contribute to symptoms [34].

Previous running-related injury (RRI) was associated with higher odds for postpartum running-related pain in the current study. To our knowledge, only one previous study reported an association between pain during pregnancy and pain in postpartum runners; however, it was not clear if this previous pain was running-related [10]. A history of a previous injury has been well established as a strong risk factor for future RRI injury in numerous prospective studies investigating non-postpartum populations [4,35–39]. Whereas previous injury may be a non-modifiable risk factor, improved rehabilitation programs may assist reducing any subsequent injuries [38]. Although the relationship between previous and subsequent RRI has not been studied in any population, clinicians may want to screen postpartum runners with previous RRI to identify running-related risk factors as well as increase education on training related risk factors, to decrease pain and future injury in this population [35].

Women with incontinence were found to have increased odds of postpartum running-related pain. Nevertheless, from the results of the present study, it is unknown if the pain caused the incontinence, or the incontinence caused the pain. Pain, specifically low back pain, has been associated with urinary incontinence in large epidemiological studies [40]. Thirty percent of postpartum mothers experience urinary incontinence and 10% experience anal incontinence [41] as pregnancy and parity are well known causes of pelvic floor dysfunction [42]. Participating in a sport, specifically one with high impact such as running, is also a risk factor for incontinence [42]. This finding of an association between incontinence and pain suggests that screening for incontinence in postpartum runners should be routinely performed and that appropriate referrals to pelvic health physical therapists are recommended.

A lack of sleep was also associated with postpartum running-related pain in this study. Sleep deprivation can increase the prevalence of clinical pain and change pain processing [43,44]. Chronic insufficiency of sleep can lead to sensitization and habituation [45,46]. Post-partum women experience significant sleep disruptions after childbirth due to infant sleep and feeding patterns [47,48]. To our knowledge, no studies have investigated the effects of postpartum sleep deprivation and its relationship with performance, recovery, or injury. Nonetheless, adults need between 7 and 9 hours of sleep per night for optimal health, with athletes requiring 9–10 hours per night for optimal performance [49–51]. Sleep deprivation affects pain facilitating agents and the immune system and can hinder muscle recovery and repair of damage when exercising at high intensity [52–54]. It is important for sleep to be screened [55] and postpartum runners should be educated on worsening sleep patterns and strategies to prevent sleep related problems [55].

Mode of delivery, specifically vaginal delivery, increased the odds of pain in postpartum runners. In other studies, 74.9% of the postpartum runners with pain reported a vaginal delivery [10]. To our knowledge, no other studies have investigated delivery type and running-related pain in the postpartum population. Chronic pain intensity has been observed to be higher after vaginal delivery than caesarean delivery, severely affecting mood and quality of life [56–58]. Although we did not query participants about vaginal tearing in this survey, it may be that the participants with vaginal delivery also had significant perineal trauma, which has been linked with persistent postpartum pain [59]. The findings of this study highlight that when evaluating a postpartum runner with running-related pain, questions about delivery should be routinely asked due to the potential contribution to pain intensity and potential recovery.

## Limitations

The study design is cross sectional, and correlational, consequently only non-causal associations can be inferred from the findings. Survey results are subjective to recall bias and to address this concern, analysis was restricted to three years postpartum, and runners were asked about their symptoms currently or in the past week. It is possible that this study is not a representative sample of the full postpartum running population (e.g., our sample was predominantly white, with a higher level of education). Survey methodology is limited in that it cannot collect data on possible biomechanical, musculoskeletal, and physiologic impairments that might be measured in a clinical or laboratory environment, however it is a first step towards identifying possible risk factors for running-related pain in postpartum women. There are limitations with the methodology we used to develop the clinical decision tool, methods that are traditionally used to develop clinical prediction rules [60,61]. However, these methods best reflected our purpose of combining parsimonious factors related to pain. A factor analysis or a cluster analysis could have been used, but these models do not reflect the variables associated with pain, they only reflect variables that have similar constructs (independent of pain). Further, as clinical prediction rules are generally developed from longitudinal modeling, our population sample would need to be followed over time to establish evidence for these factors as predictors of pain and clinicians should use caution when considering application of this evidence in practice. Finally, we did not measure intensity of pain in this study but merely if runners had pain or not.

## Conclusion

This study created a clinical decision tool that identified the cumulative effect of six risk factors (runner type-novice, postpartum accumulated fatigue scale score (>19), previous running injury, most recent delivery-vaginal, incontinence and amount of sleep (<6.8 hours) that were

associated with pain when running. With this information, clinicians can monitor and educate postpartum runners initiating or returning to running. Coaches can adapt running workload accounting for fatigue and sleep fluctuations to optimize recovery and performance. Future longitudinal studies that follow asymptomatic postpartum women returning to running after childbirth over time should be performed to validate these findings.

## Supporting information

**S1 File. Qualtrics survey seen by participants.**
(DOCX)

**S2 File. Anonymized data set.**
(XLSX)

## Acknowledgments

The authors would like to thank experts in the postpartum running rehabilitation field, who provided edits during survey creation and the pilot participants who provided feedback on the survey. Also, thank you to Rita Deering PT, PhD for providing expert feedback on early drafts of the manuscript.

## Author Contributions

**Conceptualization:** Shefali Mathur Christopher, Suzanne J. Snodgrass.

**Data curation:** Shefali Mathur Christopher, Chad E. Cook.

**Formal analysis:** Shefali Mathur Christopher, Chad E. Cook.

**Investigation:** Shefali Mathur Christopher.

**Methodology:** Shefali Mathur Christopher, Chad E. Cook, Suzanne J. Snodgrass.

**Project administration:** Shefali Mathur Christopher.

**Software:** Shefali Mathur Christopher.

**Supervision:** Chad E. Cook, Suzanne J. Snodgrass.

**Validation:** Suzanne J. Snodgrass.

**Writing – original draft:** Shefali Mathur Christopher, Chad E. Cook, Suzanne J. Snodgrass.

**Writing – review & editing:** Shefali Mathur Christopher, Chad E. Cook, Suzanne J. Snodgrass.

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
