## [Decision Letter · Decision Letter 0]

21 Jun 2021

PONE-D-21-18398

What are the biopsychosocial risk factors associated with pain in postpartum runners? Development of a clinical decision tool

PLOS ONE

Dear Dr. Christopher,

Thank you for submitting your manuscript to PLOS ONE. After careful consideration, we feel that it has merit but does not fully meet PLOS ONE’s publication criteria as it currently stands. Therefore, we invite you to submit a revised version of the manuscript that addresses the points raised during the review process. 

I have read each reviewers comments and I believe their suggestions made will improve the manuscript. Please address these recommendations and clarify where needed. I am looking forward to reading the revised paper.

We look forward to receiving your revised manuscript.

Kind regards,

Chris Harnish, PhD

Academic Editor

PLOS ONE

Journal Requirements:

Reviewers' comments:

Reviewer's Responses to Questions

**Comments to the Author**

1. Is the manuscript technically sound, and do the data support the conclusions?

Reviewer #1: Yes

Reviewer #2: Yes

2. Has the statistical analysis been performed appropriately and rigorously? 

Reviewer #1: I Don't Know

Reviewer #2: Yes

3. Have the authors made all data underlying the findings in their manuscript fully available?

Reviewer #1: Yes

Reviewer #2: Yes

4. Is the manuscript presented in an intelligible fashion and written in standard English?

Reviewer #1: Yes

Reviewer #2: Yes

5. Review Comments to the Author

Reviewer #1: This manuscript is well written and clearly aims to fill a void within the running/postpartum literature.

Please consider adding more information to the CONSORT diagram to include a breakdown of those with/without postpartum pain (both included and excluded in the final analysis); number of participants with missing data that were included in the analysis.

Please identify which test was used for each comparison in Table #1 (t-test or Chi-square).

Please identify if the researchers were blinded to the presence of pain among the participants during development of the tool.

I would like to first point out that I am not an expert in clinical prediction rule methodologies; however, the methods seem to utilize classical techniques. Furthermore, I suspect the target audiences will likely mirror my experience in this area. Methodological citations are largely missing. These are necessary to increase clarity and to ensure reproducibility. Please add these to justify the data cleaning/conversion methods, the bivariate and multivariate regression methods, as well as the internal validation methods employed in this manuscript.

In addition to adding citations in the "Statistical Analysis for Data Modeling" section, I would suggest a brief statement(s) speculating the impact of using other CPR methods on the final model (particularly if different factor reduction techniques were utilized).

Please clarify the "Bivariate Logistic Regression Modeling" subsection. It seems as if both 19 individual tests were conducted along with a mystery number of construct tests were conducted to assist with factor reduction. Did the results from construct tests reduce the total number of individual tests? Was only 1 variable from a construct included in the multivariate test?

Please consider varying more of the diction and syntax used in the "Statistical Analysis for Data Modeling" section as it draws heavily from citation #23.

Biomechanical, musculoskeletal, and physiologic impairments are directly identified in the "Introduction" and "Abstract" as contributors to postpartum pain with running. The authors even detail a recent Delphi study (citation #13) that considered such factors. However, the survey created and administered for this manuscript seems to only graze the surface of the potential impact that biomechanical, musculoskeletal, and physiologic impairments may have. The lack of direct biomechanical, musculoskeletal, and/or physiologic measurements included in the initial model should be directly addressed.

Reviewer #2: Overall, this is a clear, concise, and well-written manuscript. The introduction is relevant and theory based. Information about the previous study findings is presented for readers to follow the present study rationale and procedures. This is a topic with little exploration and/or previous research into identifying risk factors associated with pain in postpartum runners. The methods are generally appropriate, although clarification of a few details should be provided. At the start of the methods section, it states that the survey was administered between December 2019 and January 2021 (line 103-104), later on in the paragraph regarding study variables, it states survey was modified due to COVID in April 2021 (line 137). Please verify and adjust the dates of study collection.

Eligible participants could have had a child within three years, was the data looked at regarding when they started to have pain; did they have complications with delivery (not just tears) or recovery after childbirth, did the pain come closer to due delivery or pain when more miles were accrued, did they have pain just this time engaging in physical activity or after second or third child? I know from experience that starting to run a few weeks to months after childbirth vs. three years later is a big difference ( from both a fatigue, physically and emotionally). Moreover, where they cleared to engage in high intensity physical activity by OB/GYN.

In addition, it might have been helpful to ask if they were now running most of the miles pushing a stroller or without. Many mothers will run with a stroller and this can alter their body mechanics which can lead to injuries as well.

Using PAFS and EDPS was a great capture of data in the surveys. Statistical analysis and handle of missing data was appropriate and explained well.

Line 193- cite the calculator website in the reference section instead of in the body of the manuscript.

Overall, the results are clear and compelling. In Table 1 Columns three and four, believe it would be better to have n (%) only. No Mean +/- SD are represented in these columns. Make sure that the titles are fulling seen in rows (it could just be the way it printed out for this reviewer) but the words are cut off onto another row and hard to read.

Interesting to see the COVID- 19 data added to this manuscript. Would have been helpful to see if childcare (lack of daycare and schools) or jobs impacted as well.

In results and in the discussion, it was noted that most recent delivery – vaginal was a factor. This should be explored more. As stated in discussion, did not ask about tears or other complications. Would have been interesting to see what prior injuries the runners had previous.

Interesting sentence at end regarding pain. Pain is subjective. Many runners/ athletes may not state they have pain because they use to higher levels of discomfort.

The authors makes a contribution to the research literature in this area of investigation. Overall, this is a high quality manuscript that has implications for the theoretical basis, development, and important clinical decisions.

I believe this article should be accepted after the minor revisions are noted above. This reviewer is looking forward to reading more about this topic in the future.

6. PLOS authors have the option to publish the peer review history of their article (what does this mean?). If published, this will include your full peer review and any attached files.

Reviewer #1: No

Reviewer #2: No

---

## [Author Response · Author response to Decision Letter 0]

29 Jun 2021

PONE-D-21-18398

What are the biopsychosocial risk factors associated with pain in postpartum runners? Development of a clinical decision tool

PLOS ONE

(Author comments: We’d like to thank the review team for such a quick, thorough, and relevant review. We understand that finding good reviewers these days is very challenging for journals and we appreciate the time you’ve dedicated to our paper. Please note that line numbers and page numbers correspond to revised manuscript with track changes.)

Review Comments to the Author: Reviewer 1

Reviewer #1: This manuscript is well written and clearly aims to fill a void within the running/postpartum literature.

(Author comments: Thank you for your time to review this manuscript and provide valuable feedback!)

• Please consider adding more information to the CONSORT diagram to include a breakdown of those with/without postpartum pain (both included and excluded in the final analysis); number of participants with missing data that were included in the analysis.

(Author comments: Thank you for your edits we have now added more information to a flow diagram -See below for track changes, a clean copy was submitted. As we imputed data for the analysis (Table 2 and 3) and accounted for it in Table 1, we did not include missing values in the revised figure. Of the missing data there were 12 (0.10%), reflected in a total of 6 (1.11%) cases and Little’s test for missingness showed this missing date were missing completely at random.) 

• Please identify which test was used for each comparison in Table #1 (t-test or Chi-square) 

(Author comments: We have added a symbol and a footnote for clarity to identify when a Chi-square test was used.)

• Please identify if the researchers were blinded to the presence of pain among the participants during development of the tool.

(Author comments: Yes, researchers were blinded to the presence of pain during survey development. The pilot participant data was not included in the analysis).

• I would like to first point out that I am not an expert in clinical prediction rule methodologies; however, the methods seem to utilize classical techniques. Furthermore, I suspect the target audiences will likely mirror my experience in this area. Methodological citations are largely missing. These are necessary to increase clarity and to ensure reproducibility. Please add these to justify the data cleaning/conversion methods, the bivariate and multivariate regression methods, as well as the internal validation methods employed in this manuscript.

(Author comments: We appreciate the comment. We have added a few more references that explain the “classic” methodology used in developing this tool. Our team has actually created a number of these methods previously and have endeavored to explain this process in greater detail.)

• In addition to adding citations in the "Statistical Analysis for Data Modeling" section, I would suggest a brief statement(s) speculating the impact of using other CPR methods on the final model (particularly if different factor reduction techniques were utilized).

(Author Comments: We added additional discussion about the pros in cons in the limitations and mentioned other models that could have been considered.

Line 355, page 21: There are limitations with the methodology we used to develop the clinical decision tool, methods that are traditionally used to develop clinical prediction rules [61,62]. However, these methods best reflected our purpose of combining parsimonious factors related to pain. A factor analysis or a cluster analysis could have been used, but these models do not reflect the variables associated with pain, they only reflect variables that have similar constructs (independent of pain). Further, as clinical prediction rules are generally developed from longitudinal modeling, our population sample would need to be followed over time to establish evidence for these factors as predictors of pain and clinicians should use caution when considering application of this evidence in practice.)

• Please clarify the "Bivariate Logistic Regression Modeling" subsection. It seems as if both 19 individual tests were conducted along with a mystery number of construct tests were conducted to assist with factor reduction. Did the results from construct tests reduce the total number of individual tests? Was only 1 variable from a construct included in the multivariate test?

(Author comments: There were only two constructs which required selection of single variables and those were fatigue and sleep. To make it more clear in the paper we have added text to explain the selection process for these two constructs. 

Line 175 page 9: When there were multiple variables that measured the same construct (e.g., fatigue and PAFS score or sleep average hours and number of sleep interruptions) the authors identified the single item or scale that most accurately reflected the latent construct. For example, PAFS total score was used for fatigue and for sleep average hours was used for sleep.) 

• Please consider varying more of the diction and syntax used in the "Statistical Analysis for Data Modeling" section as it draws heavily from citation #23.

(Author comments: The author who did the analysis for this manuscript is on citation 23 and used the standard procedures associated with all mentioned statistical analysis. This section has been modified to change syntax (page 9-10)) 

• Biomechanical, musculoskeletal, and physiologic impairments are directly identified in the "Introduction" and "Abstract" as contributors to postpartum pain with running. The authors even detail a recent Delphi study (citation #13) that considered such factors. However, the survey created and administered for this manuscript seems to only graze the surface of the potential impact that biomechanical, musculoskeletal, and physiologic impairments may have. The lack of direct biomechanical, musculoskeletal, and/or physiologic measurements included in the initial model should be directly addressed.

(Author Response: Prior to this survey biomechanical, musculoskeletal and physiologic impairments had not been measured in postpartum runners. Studies have mainly focused on general running populations, or postpartum participants who are non-runners to generate information for this unique population. Even the Delphi used expert clinicians working with this population to identify variables that are present in postpartum runners and did not query postpartum runners themselves. To identify which variables were present in postpartum runners with and without pain, this survey was a first attempt at collecting information in this population. Survey methodology is limited, and it cannot collect on all possible biomechanical, musculoskeletal and physiologic data. Our lab is working on biomechanical and physiologic data collections to supplement this body of literature in the next few years.

We have edited the manuscript to add a limitation related to this point:

Line 352 page 21: Survey methodology is limited in that it cannot collect data on possible biomechanical, musculoskeletal and physiologic impairments that might be measured in a clinical or laboratory environment, however it is a first step towards identifying possible risk factors for running-related pain in postpartum women.

(Author response: Thank you for your helpful feedback. We appreciate your time and insight.)

Review Comments to the Author: Reviewer 2

Reviewer #2: Overall, this is a clear, concise, and well-written manuscript. The introduction is relevant and theory based. Information about the previous study findings is presented for readers to follow the present study rationale and procedures. This is a topic with little exploration and/or previous research into identifying risk factors associated with pain in postpartum runners. The methods are generally appropriate, although clarification of a few details should be provided. 

• At the start of the methods section, it states that the survey was administered between December 2019 and January 2021 (line 103-104), later on in the paragraph regarding study variables, it states survey was modified due to COVID in April 2021 (line 137). Please verify and adjust the dates of study collection.

(Author response: Thank you for catching this error. It should say April 2020 and has been corrected in the manuscript)

• Eligible participants could have had a child within three years, was the data looked at regarding when they started to have pain; did they have complications with delivery (not just tears) or recovery after childbirth, did the pain come closer to due delivery or pain when more miles were accrued, did they have pain just this time engaging in physical activity or after second or third child? I know from experience that starting to run a few weeks to months after childbirth vs. three years later is a big difference ( from both a fatigue, physically and emotionally). Moreover, where they cleared to engage in high intensity physical activity by OB/GYN.

(Author response: We did not collect information about when runners were cleared to resume activity because based on our experience this clearance is highly variable and dependent on the practitioner. This is directly because of the lack of information and medical guidelines for this population. However, we do report the number of weeks postpartum that participants began to run. It met the criteria for inclusion in the multivariate model but because it did not have a strong relationship with pain, was not present in the final model. This may be because a large number of factors may have influenced when they started to run and many of these would not be related to pain (e.g., when they found the time outside of mothering responsibilities and what supports they may have had for mothering). Because first time to postpartum run (in weeks) was not significant, it was not discussed or explored further. We did not ask about complicated delivery as the purpose of this survey was a first step in exploring which variables are related with postpartum pain. We do plan on further exploring this in future studies.) 

• In addition, it might have been helpful to ask if they were now running most of the miles pushing a stroller or without. Many mothers will run with a stroller and this can alter their body mechanics, which can lead to injuries as well.

(Author response: This is a good point. We did collect this information; however due to the focus of this manuscript we chose not to present the analysis of this variable in the study. We have now added it to the manuscript in Table 1 (see data below). 

Variables Total (N=538) Postpartum pain (N=176) Postpartum no pain (N=362) P value

Currently running with stroller 292 (54.2) 94 (53.4) 1 97 (54.4) 0.826

• Using PAFS and EDPS was a great capture of data in the surveys. Statistical analysis and handle of missing data was appropriate and explained well.

Line 193- cite the calculator website in the reference section instead of in the body of the manuscript.

(Author response: Thank you, we have cited the website now in the text (line 198, page 10) and reference section)

• Overall, the results are clear and compelling. In Table 1 Columns three and four, believe it would be better to have n (%) only. No Mean +/- SD are represented in these columns. 

(Author response: Thank you for helping us improve the messaging of the table. We have now been more deliberate in representing the variables that had mean and standard deviations (page 12-13). It represents AMA formatting.) 

• Make sure that the titles are fulling seen in rows (it could just be the way it printed out for this reviewer) but the words are cut off onto another row and hard to read.

(Author response: We have made sure to check this and will remember to check this in the proof as well. Thank you.)

• Interesting to see the COVID- 19 data added to this manuscript. Would have been helpful to see if childcare (lack of daycare and schools) or jobs impacted as well.

(Author response: Yes we agree. We added the question to account for change in mileage, sleep, depression etc. that would influence our understanding of pain and postpartum running. As this wasn’t a study focusing on changes in running due to COVID-19, we did not add further questions on this topic as it would have made the survey very long.)

• In results and in the discussion, it was noted that most recent delivery – vaginal was a factor. This should be explored more. As stated in discussion, did not ask about tears or other complications. Would have been interesting to see what prior injuries the runners had previous.

(Author response: Yes we agree. The mode of delivery is discussed in a paragraph on lines 335 page 21 of the paper. We agree this is an interesting topic for future research and we hope to investigate this in the future. Exploring all previous injury types was beyond the scope of this paper. As the main purpose of the paper was to identify variables associated with running-related pain, more data would have shifted the focus away from the main purpose of the current paper)

• Interesting sentence at end regarding pain. Pain is subjective. Many runners/ athletes may not state they have pain because they use to higher levels of discomfort.

(Author response: Yes we agree. And pain has multiple dimensions, which is important to capture when exploring this pain. These data would have distracted from main purpose of the current paper and were thus not included.)

• The authors makes a contribution to the research literature in this area of investigation. Overall, this is a high quality manuscript that has implications for the theoretical basis, development, and important clinical decisions.

I believe this article should be accepted after the minor revisions are noted above. This reviewer is looking forward to reading more about this topic in the future.

(Author response: Thank you for your helpful feedback. We appreciate your time and insight.)

---

## [Decision Letter · Decision Letter 1]

15 Jul 2021

What are the biopsychosocial risk factors associated with pain in postpartum runners? Development of a clinical decision tool

PONE-D-21-18398R1

Dear Dr. Christopher,

We’re pleased to inform you that your manuscript has been judged scientifically suitable for publication and will be formally accepted for publication once it meets all outstanding technical requirements.

Kind regards,

Chris Harnish, PhD

Academic Editor

PLOS ONE

Additional Editor Comments (optional):

Reviewers' comments:

Reviewer's Responses to Questions

**Comments to the Author**

1. If the authors have adequately addressed your comments raised in a previous round of review and you feel that this manuscript is now acceptable for publication, you may indicate that here to bypass the “Comments to the Author” section, enter your conflict of interest statement in the “Confidential to Editor” section, and submit your "Accept" recommendation.

Reviewer #1: All comments have been addressed

Reviewer #2: All comments have been addressed

2. Is the manuscript technically sound, and do the data support the conclusions?

Reviewer #1: Yes

Reviewer #2: Yes

3. Has the statistical analysis been performed appropriately and rigorously? 

Reviewer #1: Yes

Reviewer #2: Yes

4. Have the authors made all data underlying the findings in their manuscript fully available?

Reviewer #1: Yes

Reviewer #2: Yes

5. Is the manuscript presented in an intelligible fashion and written in standard English?

Reviewer #1: Yes

Reviewer #2: Yes

6. Review Comments to the Author

Reviewer #1: The authors have considered the questions/suggestions from both reviewers carefully and considerately; all changes made to the manuscript reflect this point.

Line 110 of the redlined copy seems uncited (I assume: Eysenbach G. Improving the Quality of Web Surveys: The Checklist for Reporting Results of Internet E-Surveys (CHERRIES). J Med Internet Res. 2004; 6(3): e34.). The correct citation should be added. Otherwise, I believe that this manuscript is ready for publication and have reflected this point in my recommendation to "accept" rather than require "minor revision."

Reviewer #2: Authors addressed comments well. No further actions are needed at this time. I believe this article is ready for publication

7. PLOS authors have the option to publish the peer review history of their article (what does this mean?). If published, this will include your full peer review and any attached files.

Reviewer #1: No

Reviewer #2: No

---

## [Editor Report · Acceptance letter]

3 Aug 2021

PONE-D-21-18398R1 

What are the biopsychosocial risk factors associated with pain in postpartum runners? Development of a clinical decision tool 

Dear Dr. Christopher:

I'm pleased to inform you that your manuscript has been deemed suitable for publication in PLOS ONE. Congratulations! Your manuscript is now with our production department. 

Kind regards, 

on behalf of

Dr. Chris Harnish 

Academic Editor

PLOS ONE